# Detection of intracranial hypertension in children using optical coherence tomography: a systematic review protocol

Sohaib R Rufai  ,[1,2] Noor ul Owase Jeelani,[3,4] Rebecca J McLean[2]

¹Clinical and Academic Department of Ophthalmology, Great Ormond Street Hospital for Children, London, United Kingdom
²University of Leicester Ulverscroft Eye Unit, Leicester Royal Infirmary, Leicester, United Kingdom
³Craniofacial Unit, Great Ormond Street Hospital for Children, London, United Kingdom
⁴Developmental Biology & Cancer Dept, UCL GOS Institute of Child Health, London, United Kingdom

**Correspondence to**
Dr Rebecca J McLean;
rjm19@leicester.ac.uk

## ABSTRACT

**Introduction** Intracranial hypertension (ICH) in children can have deleterious effects on the brain and vision. It is notoriously difficult to estimate intracranial pressure (ICP) in children and existing methods deliver suboptimal diagnostic accuracy to be used as screening tools. Optical coherence tomography (OCT) may represent a valuable, non-invasive surrogate measure of ICP, as has been demonstrated in a number of associated conditions affecting adults. More recently, OCT has been employed within the paediatric age group. However, the role of OCT in detecting ICH in children has not been rigorously assessed in a systematic review for all relevant conditions. Here, we propose a systematic review protocol to examine the role of OCT in the detection of ICH in children.

**Methods and analysis** Electronic searches in the Cochrane Central Register of Controlled Trials, Medline, Embase, Web of Science and PubMed will identify studies featuring OCT in detecting ICH in children. Two independent screeners will identify studies for inclusion using a screening questionnaire. The systematic search and screening will take place between 2 April 2020 and 1 June 2020, while we aim to complete data analysis by 1 September 2020. Quality assessment will be performed using the National Institutes of Health Quality Assessment Tool for Observational Cohort and Cross-Sectional Studies. The primary outcome measure is the sensitivity and specificity of OCT in detecting ICH in children. Secondary outcomes measures include conditions associated with ICH per study, direct ICP monitoring, sensitivity and specificity of other measures for ICP and OCT parameters used.

**Ethics and dissemination** Ethical approval is not required for the proposed systematic review as no primary data will be collected. The findings will be disseminated through presentations at scientific meetings and peer-reviewed journal publication.

**PROSPERO registration number** CRD42019154254.

## INTRODUCTION
### Intracranial hypertension

Intracranial hypertension (ICH) was first described by Quincke in 1896 and remains a subject of major importance among the medical profession.[1] ICH can pose devastating consequences if untreated, including

### Strengths and limitations of this study

► The proposed study will be the first to examine the role of optical coherence tomography for all paediatric conditions associated with intracranial hypertension.
► This protocol provides transparency to the proposed systematic review methodology and reduces the possibility of duplication.
► The proposed methodology is recommended by the Cochrane Collaboration.
► A scarcity of relevant randomised controlled trials and associated high risk of bias of observational studies may be the main limitations in the evidence returned by this systematic review.
► The broad search strategy may result in a large number of titles and abstracts to screen initially.

visual impairment, neurocognitive delay, disability and death.[2 3] Subacute conditions in children can cause insidious elevation of intracranial pressure (ICP) which may go undetected before insult to the brain and vision has already begun.[2 4] Hence, early detection of ICH is crucial in enabling timely medical intervention to prevent or limit the sequelae of ICH.

ICH affects between 0.63 and 0.71 per 100 000 children.[5 6] ICH can be broadly classified as primary (or idiopathic) and secondary. Primary ICH is characterised by elevated ICP with no identifiable cause.[4 7] Common risk factors for primary ICH include female gender, obesity and postpubertal status, although it is known to occur in children with a lack of predilection for gender and obesity.[4 8–14] Secondary ICH describes elevated ICP that is a direct result of another condition, for example, an expansive intracranial process, such as space occupying lesion or hydrocephalus, or a constricting skull pathology, such as the premature fusion of cranial sutures in craniosynostosis.[2–4]

## Measurement of ICP

ICP measurement in children is notoriously difficult. The gold standard measure is direct intraparenchymal ICP measurement, which involves overnight admission to hospital and general anaesthesia. The risks of intraparenchymal ICP measurement include infection, haemorrhage, cerebrospinal fluid (CSF) leak and device failure.[15 16] Furthermore, there is no universal agreement on timing, duration and frequency for accurate measurement or indeed agreement on the normal paediatric range of ICP – 11 to 15 mm Hg is widely considered normal, but Hayward *et al* argue that the upper limit could reasonably be raised to 20 mm Hg.[17 18]

The ideal screening method for ICH should be sensitive, specific and child friendly with the ability to record objective serial measurements over time. However, existing methods fail to satisfy all of these criteria. Visual evoked potentials (VEP) assess the amplitude and latency time of the averaged encephalographic response to visual stimulus. Axonal injury secondary to ICH is associated with reduction of amplitude or prolongation of the latency time.[2 19] However, limitations of VEP include the need for good cooperation by the child, plus high variability in normal subjects.[20] Fundus examination can permit direct visualisation of papilloedema or optic atrophy, both of which are associated with ICH. A study by Tuite *et al* demonstrated that fundus examination had sensitivity of 100% in detecting ICH in children aged over 8 years, but only 22% for that in younger patients, excluding it as a suitable screening method. B-scan ocular ultrasound can be helpful in acute situations for detecting severe ICH, but possesses a sensitivity of only 11% when compared with funduscopy.[21–24] Radiological signs, such as the 'beaten copper' appearance of the cranium, universally demonstrate poor sensitivity for ICH.[25]

## Optical coherence tomography

The optic nerve is primarily intracranial. Thus, ICH causes optic nerve head swelling, termed papilloedema, with secondary retinal changes. These changes can be measured objectively using optical coherence tomography (OCT)—a non-invasive imaging technique that can acquire ultrahigh resolution, three-dimensional images of the back of the eye within seconds.[26] OCT has been used to describe the normal development of the optic nerve[27] and fovea[27 28] in children. OCT has also been used in a wide range of conditions within paediatric ophthalmology, including retinopathy of prematurity,[29] retinoblastoma,[30] nystagmus,[31] albinism,[32] achromatopsia,[33] foveal hypoplasia,[34] optic nerve hypoplasia,[35] primary congenital glaucoma,[36] microcephaly[37] and others. Furthermore, OCT has been successfully used as a diagnostic tool in a variety of conditions associated with ICH, including idiopathic ICH, craniosynostosis and hydrocephalus.[38–44]

Thus, OCT may represent an ideal screening method for ICH in children owing to its quick acquisition time and non-invasive, child-friendly nature. However, this has not yet been rigorously assessed in a systematic review. Here, we propose to conduct the first systematic review examining the role of OCT in all conditions associated with ICH in children.

## Objectives

The primary research objective of this proposed systematic review is to assess the sensitivity and specificity of OCT in detecting ICH in children.

The secondary research objectives are as follows:
► To assess which conditions associated with ICH have been successfully studied by OCT.
► To assess OCT success rate per condition per study.
► To identify surrogate measures of ICP and their associated sensitivity, specificity and success rates.
► To assess which ICP range is determined as normal per condition per study.

## METHODS

This protocol has adhered to the Preferred Reporting Items for Systematic Review and Meta-Analysis Protocols (PRISMA-P) checklist.[45]

### Eligibility criteria

The Population, Intervention, Comparison, Outcome and Study design strategy[46] has been employed to outline the eligibility criteria for this systematic review, as summarised in table 1.

### Types of outcome measures

The main outcome measure will be sensitivity and specificity for OCT measures in detecting ICH. OCT metrics may include one or more of the following:
► Optic nerve parameters: cup depth, cup width, cup volume, disc width and cup to disc ratio.
► Rim parameters: retinal nerve fibre layer thickness, rim area, rim volume, Bruch's membrane opening-minimum rim width, Bruch's membrane orientation, ganglion cell layer thickness, full peripapillary analysis.
► Retinal parameters: macular and perimacular retinal thickness, foveal pit width, depth and area, plus segmentation of all retinal layers.

The secondary outcome measures are as follows:
► Condition(s) associated with ICH per study.
► OCT success rate.
► Other surrogate estimates of ICP.
► Other OCT parameters not listed above.
► ICP range determined as normal.

### Quality-of-life outcomes

Quality-of-life outcomes and patient satisfaction measured by surveys will be included where applicable.

### Adverse events

The process of OCT imaging cannot directly produce adverse events as it is a safe and non-invasive imaging method that uses low-coherence light.[47] It does not involve ionising radiation.

**Table 1** Eligibility criteria

| PICOS strategy[46] | Inclusion criteria | Exclusion criteria |
|---|---|---|
| P—Population | Studies of children, defined as being aged under 18 years, diagnosed with conditions associated with ICH. | (i) Studies of adults aged 18 years old or over; (ii) Studies not pertaining to the diagnosis of ICH. |
| I—Intervention | Studies employing OCT to detect ICH. | Any studies that do not include OCT. |
| C—Comparator | Absence of a comparator will not lead to exclusion of studies, as it may be unethical to deprive one arm of the study of OCT when it may lead to a better clinical outcome in such a dangerous situation of ICH. | N/A |
| O—Outcome | Sensitivity and specificity of any OCT measure(s), ±surrogate measure(s), for ICH. | Studies that do not report OCT measures. |
| S—Study design | All level IV evidence and above, that is, systematic reviews, randomised controlled trials, cohort studies, case-control studies and case series, as defined by the Oxford Centre for Evidence-based Medicine.[51] | Level 5 evidence, that is, expert opinion without critical appraisal. |

ICP, intracranial pressure; N/A, Not applicable; OCT, optical coherence tomography; PICOS, Population Intervention Comparison Outcome and Study design.

### Follow-up

There will be no minimum follow-up period set for study inclusion. In many cases, the decision to employ rapid intervention (ie, surgery) is made as soon as possible once ICH has been diagnosed. The stipulation of a minimum follow-up period would therefore risk losing valuable data from excluded studies.

### Information sources
#### Record characteristics

There shall be no time restriction on records considered for this study. Unpublished records shall be considered. There shall be no language restrictions; all applicable non-English records will undergo professional translation from the University of Leicester Centre for Translation and Interpreting Studies.

#### Electronic searches

The following platforms shall be searched:
► Cochrane Central Register of Controlled Trials (including the Cochrane Eyes and Vision Group Trials Register).
► EMBASE Classic+Embase (1947 to present).
► Ovid MEDLINE(R) (1946 to present).
► Ovid MEDLIN In-Process and Other Non-Indexed Citations.
► PubMed (1948 to present).
► Web of Science Core Collection (1970 to present).

#### Searching other resources

Unpublished studies identified by experts will be considered against our inclusion criteria. References of included studies will be searched and the authors contacted by email or letter to identify further unpublished works where applicable. If the authors do not reply within 14 days, a reminder will be sent by email or letter. If the authors do not reply to the reminder within 14 days, their study will be excluded.

### Search strategy

Medical subject headings terms for 'intracranial hypertension' and 'optical coherence tomography' were entered into the above search platforms. Full details of search terms and strategy are included in online supplementary appendix 1.

### Study records
#### Data management

EndNote V.X9 (Thomson Reuters, New York, New York, USA) reference management software will be used for data management.

#### Selection of studies

Two independent screeners (SRR and RJM) shall follow a three-stage screening method. The search will be conducted on 2 April 2020 and screening will be performed until 1 June 2020. Online supplementary appendix 2 outlines the screening questions and process. First, titles will be screened, followed by abstracts, followed by full papers. At each stage, studies shall be classified as 'include', 'exclude' or 'unclear', with those not excluded progressing to the next stage of screening. If any studies are classified as 'unclear' following the full paper screen, the study authors in question would be contacted by email or letter for clarification. If the authors fail to reply within 14 days, a reminder will be sent. If the authors fail to reply to the reminder within 14 days, their study will be excluded. If there is any disagreement over which papers

| Table 2 | Data to be extracted |
|---|---|
| **Data category** | **Data to be extracted** |
| Study characteristics | ▶ Author(s)<br>▶ Year<br>▶ Study design<br>▶ Study location<br>▶ Number of patients<br>▶ Mean age<br>▶ Age range |
| Primary outcome measures | ▶ OCT parameter(s)<br>▶ Sensitivity (%)<br>▶ Specificity (%) |
| Secondary outcome measures, where available | ▶ Condition(s) studies;<br>▶ OCT success rate;<br>▶ Other surrogate estimates of ICP;<br>▶ Sensitivity (%)<br>▶ Specificity (%)<br>▶ Other OCT parameters not listed above;<br>▶ ICP range determined as normal.<br>▶ Quality-of-life outcomes<br>▶ Adverse events |

ICP, intracranial pressure; OCT, optical coherence tomography.

are selected for inclusion, the third arbitrator (NuOJ) shall make the final decision.

### Data collection process
Our data extraction tool is included in online supplementary appendix 3, which is adapted from the Cochrane Collaboration.[48]

### Data items, outcomes and prioritisation
Table 2 outlines the data items to be extracted. Extraction of primary outcome measures will be addressed as these will be used to address the primary research objective of this systematic review.

### Risk of bias in individual studies
The National Institutes of Health (NIH) Quality Assessment Tool for Observational Cohort and Cross-Sectional Studies,[49] included in online supplementary appendix 4, shall be used to assess risk of bias in the included studies.

### Data analysis
Scoping searches suggest that mainly observational studies will be returned by our search strategy with few relevant RCTs. Weighted means for outcome measures will only be calculated if multiple RCTs are identified. Otherwise, we shall perform a qualitative review summarising the available evidence. A narrative synthesis will be

conducted, examining the strengths, weaknesses, similarities and differences between included studies. Outcomes measures will be extracted and reported descriptively. Included studies will be evaluated based on the strength of the evidence. We aim to complete data analysis by 1 September 2020.

### Confidence in cumulative evidence
The Grades of Recommendation, Assessment, Development, and Evaluation (GRADE) approach[50] will be employed to evaluate the strength of the body of evidence identified in this study, where applicable.

### Patient and public involvement
Involvement of patients and the public in the development of this systematic protocol is not applicable, as no patient recruitment will take place.

## ETHICS AND DISSEMINATION
Ethical approval is not required for the proposed systematic review as no primary data will be collected. The findings of this systematic review shall be disseminated through presentations at scientific meetings, as well as peer-reviewed, open-access journal publication. Data generated during this systematic review will be made available from the corresponding author on reasonable request.

## DISCUSSION
To the best of our knowledge, this is the first proposed systematic review to rigorously examine the role of OCT in detecting ICH in children. We have adhered to the PRISMA-P checklist[45] when developing this protocol. Two independent screeners shall conduct the search, with a third arbitrator available if required. Quality will be assessed using the NIH Quality Assessment Tool for Observational Cohort and Cross-Sectional Studies.[49] This systematic review should help to clarify the role of OCT as a non-invasive screening method for ICH in paediatrics. The main limitations of the study may be due to the low number of relevant RCTs and higher risk of bias associated with observational studies returned by the systematic review. Publication of this protocol should provide transparency to our proposed systematic review methodology as well as reduce the possibility of duplication.

**Acknowledgements** The authors would like to thank Irene Gottlob and Frank Proudlock of the University of Leicester Ulverscroft Eye Unit, for their support and contributions towards this systematic review protocol.

**Collaborators** N/A.

**Contributors** SRR: concept, methodology, protocol writing and final approval. NuOJ: supervision, concept, critical revision and final approval. RJM: supervision, methodology, critical revision, final approval and guarantor of review.

**Funding** The Medical Research Council (London, UK; grant no.: MR/N004566/1). SRR is funded by a National Institute for Health Research (NIHR) Doctoral Fellowship for this research project. This paper presents independent research funded by the

NIHR and MRC. The views expressed are those of the authors and not necessarily those of the MRC, the NHS, the NIHR or the Department of Health and Social Care.

**Disclaimer** The views expressed are those of the author(s) and not necessarily those of the MRC, the NHS, the NIHR or the Department of Health and Social Care.

**Competing interests** None declared.

**Patient and public involvement** Patients and/or the public were not involved in the design, or conduct, or reporting, or dissemination plans of this research.

**Patient consent for publication** Not required.

**Provenance and peer review** Not commissioned; externally peer reviewed.

**Open access** This is an open access article distributed in accordance with the Creative Commons Attribution 4.0 Unported (CC BY 4.0) license, which permits others to copy, redistribute, remix, transform and build upon this work for any purpose, provided the original work is properly cited, a link to the licence is given, and indication of whether changes were made. See: https://creativecommons.org/licenses/by/4.0/.

**ORCID iD**
Sohaib R Rufai http://orcid.org/0000-0001-8134-6393

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
