## [Reviewer comments · BMJ Open]

ARTICLE DETAILS

TITLE (PROVISIONAL)	Detection of Intracranial Hypertension in Children using Optical Coherence Tomography: A Systematic Review Protocol
AUTHORS	Rufai, Sohaib; Jeelani, Noor ul Owase; McLean, Rebecca

VERSION 1 - REVIEW

REVIEWER	Ana Banc "Iuliu Hatieganu" University of Medicine and Pharmacy Cluj-Napoca, Romania
REVIEW RETURNED	28-Feb-2020

GENERAL COMMENTS	The manuscript is very well written and the subject of this systematic review has a high clinical relevance. I have two suggestions for the authors: 1). Please provide the dates of the study in this study protocol (or at least, the tentative dates) - to prove that the study is planned or ongoing; 2). In Table 1 you defined the study population "as being aged under 18 years", whereas in Appendix 2 you defined the participants "as 0 to 16 years of age". It should be the same definition throughout the protocol.
---

REVIEWER	Rita Gama Gama Eye Care and Hospital da Luz, Lisbon, Portugal
REVIEW RETURNED	12-Mar-2020

GENERAL COMMENTS	In page 7, lines 22-29: I suggest you include the sensitivity and specificity of the following OCT parameters: the cup volume, rim area and ganglion cell layer thickness (evaluated by other OCT software's).
--

REVIEWER	Aristotelis Kalyvas Division of Neurosurgery, University Health Network, University of Toronto, Toronto, Ontario, Canada
REVIEW RETURNED	11-Apr-2020

GENERAL COMMENTS	The authors have presented a systematic review protocol that aims at detection of Intracranial Hypertension in Children using optical coherence tomography. I believe that the protocol addresses the background of the condition, the objective of the study, the gap of knowledge and the rationale for the study in a systematic and sound way. The methodology seems solid and the description of the outcomes detailed. The inclusion/exclusion criteria have been described nicely according to PICOS strategy. However, there is a minor discrepancy between the data in the "Table 1" and the "Appendix 2: Screening questions" that the authors need to amend. Specifically, in "Table 1" studies that include participants below 18 years of age can be considered for inclusion, while in the "Appendix 2: Screening questions" the pertinent age threshold is 16 years of age. Moreover, in the "strengths and limitations of the study" the authors have aptly addressed the strengths of the study, however, I cannot see any mentioning of the anticipated difficulties and limitations of the study. I suggest that the authors can add a few points covering this aspect in their revision. Finally, there is a grammatical/typing error: "evaluated" instead of "evaluate" in page page 10, line 6. Overall, this is an interesting topic and a very well written protocol.
---

VERSION 1 – AUTHOR RESPONSE

Suggestion, Question, or Comment from Reviewer 1: Ana Banc	Author's Response	Manuscript section featuring changes
The manuscript is very well written and the subject of this systematic review has a high clinical relevance. I have two suggestions for the authors: 1). Please provide the dates of the study in this study protocol (or at least, the tentative dates) - to prove that the study is planned or ongoing;	We wish to thank the Reviewer for this positive feedback. We have included the proposed dates of the study in the Abstract and Methods sections: The systematic search and screening will take place between 2nd April 2020 and 1st June 2020, whilst we aim to complete data analysis by 1st September 2020. Please note that we have not yet commenced the screening process.	Abstract; Methods: Selection of studies; Methods: Data analysis.
2). In Table 1 you defined the study population "as being aged under 18 years", whereas in Appendix 2 you defined the participants "as 0 to 16 years of age". It should be the same definition throughout the protocol.	Many thanks for highlighting this – we have amended Appendix 2 to the definition "aged under 18 years".	Appendix 2

Suggestion, Question,	Author's Response	Manuscript section featuring
-------------------	------------------------------

or Comment from Reviewer 2: Rita Gama		changes
In page 7, lines 22-29: I suggest you include the sensitivity and specificity of the following OCT parameters: the cup volume, rim area and ganglion cell layer thickness (evaluated by other OCT software's).	Many thanks, we have included your suggested OCT parameters.	Methods: types of outcome measures

Suggestion, Question, or Comment from the Reviewer 3: Aristotelis Kalyvas	Author's Response	Manuscript section featuring changes
The authors have presented a systematic review protocol that aims at detection of Intracranial Hypertension in Children using optical coherence tomography. I believe that the protocol addresses the background of the condition, the objective of the study, the gap of knowledge and the rationale for the study in a systematic and sound way. The methodology seems solid and the description of the outcomes detailed. The inclusion/exclusion criteria have been described nicely according to PICOS strategy.	We wish to thank the Reviewer for this positive feedback.	N/A
However, there is a minor discrepancy between the data in the "Table 1" and the "Appendix 2: Screening questions" that the authors need to amend. Specifically, in "Table 1" studies that include participants below 18 years of age can be considered for inclusion, while in the "Appendix 2: Screening questions" the pertinent age threshold is 16 years of age.	Many thanks for highlighting this – we have amended Appendix 2 to the definition “aged under 18 years”.	Appendix 2
Moreover, in the "strengths and limitations of the study" the authors have aptly addressed the strengths of the study, however, I cannot see any mentioning of the anticipated difficulties and limitations of the study. I suggest that the authors can add a few points covering this aspect in their revision.	Many thanks – we have included the following anticipated limitations: “A scarcity of relevant randomised controlled trials and associated high risk of bias of observational studies may be the main limitations in the evidence returned by this systematic review.	Strengths and limitations Discussion

	The broad search strategy may result in a large number of titles and abstracts to screen initially.”	
Finally, there is a grammatical/typing error: "evaluated" instead of "evaluate" in page page 10, line 6. Overall, this is an interesting topic and a very well written protocol.	Many thanks for pointing this out this error – we have amended as advised. Many thanks for the positive feedback.	Methods: confidence in cumulative evidence

VERSION 2 – REVIEW

REVIEWER	Aristotelis Kalyvas Division of Neurosurgery, Toronto Western Hospital/University Health Network, University of Toronto, Toronto, Ontario, Canada
REVIEW RETURNED	03-May-2020

GENERAL COMMENTS	The authors have presented a systematic review protocol that aims at detection of Intracranial Hypertension in Children using optical coherence tomography. I believe that the protocol addresses the background of the condition, the objective of the study, the gap of knowledge and the rationale for the study in a systematic and sound way. The methodology seems solid and the description of the outcomes detailed. The inclusion/exclusion criteria have been described nicely according to PICOS strategy. Moreover, in the "strengths and limitations of the study" the authors have aptly addressed the strengths and limitations of the study. Overall, this is an interesting topic and a very well written protocol.
--